# Effect of Quartz on the Preparation of Sodium Stannate from Cassiterite Concentrates by Soda Roasting Process

**Yuanbo Zhang, Benlai Han, Zijian Su \*, Xijun Chen, Manman Lu, Shuo Liu, Jicheng Liu and Tao Jiang**

School of Minerals Processing and Bioengineering, Central South University, Changsha 410083, China; zybcsu@126.com (Y.Z.); hblcsu@126.com (B.H.); 17786528627@163.com (X.C.); lmmcsu@163.com (M.L.); lsus91@163.com (S.L.); ljccsu@126.com (J.L.); jiangtao@csu.edu.cn (T.J.)
\* Correspondence: suzijian@csu.edu.cn; Tel.: (+86)-731-88877214

**Abstract:** Sodium stannate ($Na_2SnO_3$) has been successfully prepared by a novel process of roasting cassiterite concentrates and sodium carbonate ($Na_2CO_3$) under $CO$–$CO_2$ atmosphere, namely soda roasting-leaching process. However, more than 22 wt. % tin of the cassiterite was not converted into $Na_2SnO_3$ and entered the leach residues. Quartz ($SiO_2$) is the predominant gangue in the cassiterite, and phase evolution of $SnO_2$–$SiO_2$–$Na_2CO_3$ system roasted under $CO$–$CO_2$ atmosphere was still uncertain. In this study, the effect of $SiO_2$ in cassiterite concentrates on preparation of $Na_2SnO_3$ was clarified. The results indicated that $Na_8SnSi_6O_{18}$ was inevitably formed when cassiterite and $Na_2CO_3$ were roasted above 775 °C under $CO$–$CO_2$ atmosphere via the reaction of $SnO_2 + 6SiO_2 + 4Na_2CO_3 = Na_8SnSi_6O_{18} + 4CO_2$, and formation of $Na_8SnSi_6O_{18}$ would be increased with increasing roasting temperature and Si/Sn mole fraction. In addition, it was found that $Na_8SnSi_6O_{18}$ was insoluble in the leachate at pH value range of 1–14, which, therefore, was enriched in the leach residues. The silicon content of the cassiterite concentrates should be controlled as lower as possible to obtain a higher conversion ratio of $Na_2SnO_3$.

**Keywords:** sodium stannate; cassiterite concentrates; silicon oxide; sodium stannic silicate

## 1. Introduction

Sodium stannate ($Na_2SnO_3$) is highly desirable in many fields, including electroplating [1], tin alloy production [2], and solid superbase catalysts for dehydrogenation and flame retardants [3,4]. Recently, it is also used as solid electrolytes and electrode materials in chemical sources of electrical energy [5]. The consumption of sodium stannate has been increasing rapidly in the last decade. The traditional $Na_2SnO_3$ preparation processes applied metallic tin and low-melting-point sodium hydroxide (NaOH) as raw materials, which were conducted in a fused state in the presence of sodium nitrate ($NaNO_3$) as oxidizers [6]. However, metallic tin was always obtained from high-temperature reduction smelting process. Besides, some secondary tin-containing resources, including stanniferous alloy, tin scrap, waste solder, and electronic waste, have also been used for preparing sodium stannate [7–13], and these processes would cause high production cost and long process flow. In addition, the emission of hazardous gases ($NH_3$ and NO$x$) deriving from the oxidizers ($NaNO_3$) was also a shortcoming.

The authors' group has developed a novel process for preparing $Na_2SnO_3$ from cassiterite concentrates ($SnO_2$) and sodium carbonate ($Na_2CO_3$) roasted in a solid-state under $CO$–$CO_2$ atmosphere [14,15], namely soda roasting–leaching process. Sodium stannate trihydrate ($Na_2SnO_3$·$3H_2O$) with a purity of 95.8 wt. % was obtained, which met the requirement of industrial first-grade products.

This process displays a bright prospect in preparing $Na_2SnO_3$, and the function mechanism and the formation kinetics of $Na_2SnO_3$ have also been clarified in previous publications [16–19]. It was found that CO–$CO_2$ atmosphere promoted the formation of oxygen deficiency on the surface of cassiterite, which broke the stable structure of $SnO_2$. Then the activation energy of the reactions between $Na_2CO_3$ and $SnO_2$ decreased significantly, and the formation of $Na_2SnO_3$ was much easily under CO–$CO_2$ atmosphere [15–20]. Nevertheless, it was found that over 22 wt. % tin of the cassiterite was not converted into $Na_2SnO_3$ and disposed as residues during the roasting–leaching process [20].

Quartz ($SiO_2$) is the predominant gangue mineral in cassiterite ores, which cannot be separated perfectly by beneficiation combined methods, such as gravity concentration and froth flotation [21,22]. Before used as raw materials, the cassiterite concentrates should be firstly pretreated by oxidation roasting–acid leaching process to remove the impurity elements, including Fe, As, S, Pb, Sb, etc. [23,24]. However, the quartz ($SiO_2$) is very hard to be removed during the pretreatment process.

Our previous studies have discussed in detail about the formation mechanism of $Na_2SnO_3$; however, the reaction principle and thermodynamic data of $Na_2CO_3$–$SnO_2$–$SiO_2$ system were unclear [25–27]. Hence, the major objectives of this study were: (1) to determine the effect of $SiO_2$ on the leaching efficiency of Sn and Si; (2) to investigate the effect of $SiO_2$ on phase evolution of $SnO_2$–$Na_2CO_3$ system; (3) to ascertain leaching characteristics of the tin, silicon-containing compounds.

## 2. Materials and Methods

### 2.1. Materials

As described in our previous studies [14], firstly, cassiterite concentrates were roasted in air at 900 °C for 120 min and then leached with 25% HCl to remove the main impurity elements, including Fe, As, S, Pb, and Sb. The chemical compositions of the original and pretreated cassiterite concentrates are given in Table 1. All the testing samples were pre-ground to a particle size passing through a 200 mesh screen (<0.074 mm). The gases used in this study included CO, $CO_2$, and $N_2$ gases, all of which were with purity of 99.99 vol. %.

**Table 1.** Chemical compositions of the original and retreated cassiterite concentrates (wt. %).

| Element | Sn | Si | Fe | CaO | S | $Al_2O_3$ | Zn | As | Pb |
|---|---|---|---|---|---|---|---|---|---|
| Raw material | 42.9 | 3.9 | 8.86 | 8.31 | 5.11 | 1.16 | 1.21 | 0.50 | 0.38 |
| Pretreated | 62.9 | 3.7 | 0.11 | 0.17 | 0.04 | 0.28 | 0.02 | 0.03 | 0.03 |

Analytically pure $Na_2CO_3$, $SnO_2$, and $SiO_2$ were also used to investigate the reaction mechanism.

### 2.2. Methods

#### 2.2.1. Roasting Process

Sodium salt roasting was one of the effective methods to treat minerals [28–31]. The roasting tests were performed in the high-temperature zone of a horizontal electric resistance furnace. An experimental schematic diagram for the roasting tests is the same as that reported in the previous study [16,32]. The materials were weighed precisely at a certain molar ratio and mixed up gently with an agate mortar and pestle for 30 min. After that, mixed samples were dried in a drying oven at 105 °C for 4 h and then a dried sample about 5.0 g was placed in a corundum crucible (80 mm × 10 mm) and loaded into a heat resistant corundum tube (diameter 45 mm). The crucible carrying the sample was pushed toward the constant roasting zone located in the central area of an electrically heated horizontal tube furnace. Beforehand, $N_2$ gas was introduced into the corundum tube until the temperature reached a constant value. Next, the $N_2$ was immediately replaced by the mixed CO–$CO_2$ gas. Inlet gas flow rate was fixed at 4.0 L/min. The sample was then roasted in a 15 vol. % CO atmosphere at a given temperature (775 °C, 825° C, 875 °C, and 925 °C) for different roasting time (5 min, 10 min, 15 min,

20 min, 30 min, and 60 min). After that, the roasted samples were cooled in pure $N_2$ atmosphere. Finally, the cooled samples were ready for further analyses.

### 2.2.2. Leaching Process

The leaching tests were conducted in 250 mL round bottom flasks with a mechanical stirring paddle, and the stirring rate was fixed at 300 rpm for each test. A water bath was used to control the temperature at 40 °C. Next, the ground roasted products of 10.0 g and distilled water of 40 mL were put into the flasks and leached at a fixed pH solution for 60 min in the water bath (pH was fixed at 12.5 based on our previous study [14]). Finally, the leaching solution was filtered and prepared for the determination of Sn and Si concentration. The residues were washed with distilled water to identify the phase constituents.

The leaching efficiency of Sn and Si, which is calculated according to the following equation:

$$L = \frac{1000CV}{MW}100\% \tag{1}$$

where $L$ is the leaching efficiency of Sn or Si, $M$ is the weight of the roasted samples (g), $W$ is the Sn or Si grade of the roasted samples (%), $C$ is the mass concentration of Sn or Si in the leaching solution (mg/mL), and $V$ is the volume of leaching solution (mL).

### 2.2.3. Instrument Techniques

The chemical compositions of cassiterite concentrates were examined using an X-ray fluorescence spectrometer (XRF, Axios MAX, PANalytical, Almelo, The Netherlands). The phase constituents of the samples were identified by X-ray diffraction (XRD; D/max 2550PC, Rigaku Co. Ltd, Tokyo, Japan) with the step of 0.02° at 10°·min$^{-1}$ ranging from 10° to 80°. The content of Sn and Si in the solid material and the aqueous solution were determined using inductively coupled plasma atomic emission spectroscopy (ICP-AES; Icap7400 Radial, Thermo Fisher Scientific, Waltham, MA, USA).

## 3. Results and Discussion

### *3.1. Behaviour of Si during the Soda Roasting-Leaching Processs*

#### 3.1.1. Effect of Roasting Temperatures and Time

Our previous studies showed that trihydrate sodium stannate ($Na_2SnO_3\cdot3H_2O$) was obtained by roasting cassiterite concentrates and $Na_2CO_3$ in CO–CO$_2$ atmosphere [14], and other impurity elements were almost removed in the pretreatment process. However, the pretreated cassiterite concentrates still contained 3.66 wt. % Si. To examine the impact of $SiO_2$ on the formation of $Na_2SnO_3$, the leaching efficiency of Sn and Si was firstly investigated at different roasting temperatures and time.

Figures 1 and 2 illustrate the effect of roasting temperatures and roasting time on leaching efficiency of Sn and Si under the setting experimental conditions: CO content of 15% and $Na_2CO_3$/$SnO_2$ mole ratio of 1.5.

It was observed from Figure 1 that the roasting temperatures had significant impact on leaching efficiency of Sn and Si. The leaching efficiency of Sn and Si increased with increasing the roasting temperatures. The leaching efficiency of Sn and Si increased significantly as roasting temperatures increased from 800 °C to 1000 °C. At 1000 °C, the leaching efficiency of Sn and Si was almost the same level, 97.5 wt. % and 96.1 wt. %, respectively. The melting point of $Na_2CO_3$ is reported as 851 °C, and melted $Na_2CO_3$ is inclined to volatilize, which resulted in equipment corrosion at higher temperatures [33,34]. Therefore, 875 °C is selected to be a suitable roasting temperature based on the previous study [14].

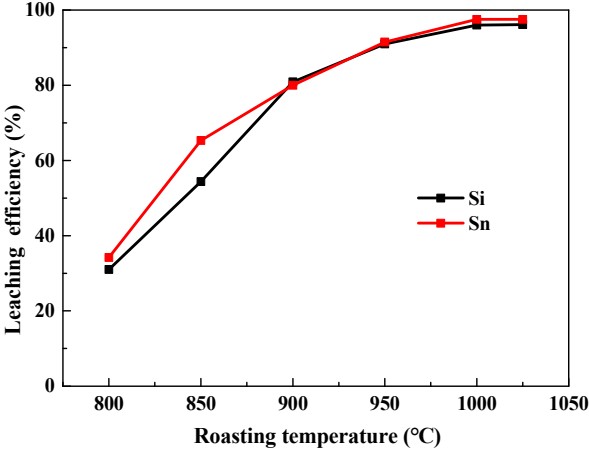

**Figure 1.** Effect of roasting temperature on leaching efficiency of Sn and Si (Time: 60 min).

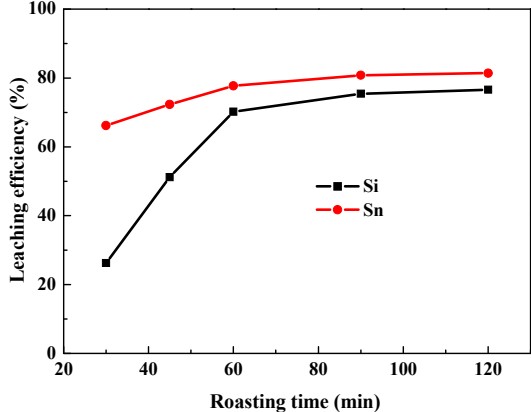

**Figure 2.** Effect of roasting time on leaching efficiency of Sn and Si (Temperature: 875 °C).

Figure 2 shows the effect of roasting time in the range of 30–120 min on the leaching efficiency of Sn and Si. The leaching efficiency of Si increased from 26.3 wt. % to 70.2 wt. % as the roasting time increased from 30 to 60 min, while the Sn leaching efficiency was higher than 65 wt. % as the roasting time prolonged. The results indicated that reaction rate of Equation (2) was much faster than that of Equation (3) in CO–CO$_2$ atmosphere.

$$SnO_2 + Na_2CO_3 = Na_2SnO_3 + CO_2 \qquad (2)$$

$$SiO_2 + Na_2CO_3 = Na_2SiO_3 + CO_2 \qquad (3)$$

When the roasting time was further extended to 90 min, the leaching efficiency of Sn and Si almost stayed unchanged. As the roasting time was 120 min, the leaching efficiency of Sn and Si was 81.4 wt. % and 76.6 wt. %, respectively.

Based on the results given in Figures 1 and 2, the distribution of Sn and Si elements in the whole experimental flowsheet is presented in Figure 3. The content of Sn and Si in pretreated cassiterite concentrates were 62.9 wt. % and 3.7 wt. %, respectively. Then, the roasted samples were leached in the setting experimental conditions. 70.2 wt. % of Si entered the leachate. Meanwhile, it was worthy to note that leaching efficiency of Sn was 77.7 wt. % and 22.3 wt. % Sn entered the residues. To discover the underlying reason, the phase components of roasted samples and leach residues should be further determined.

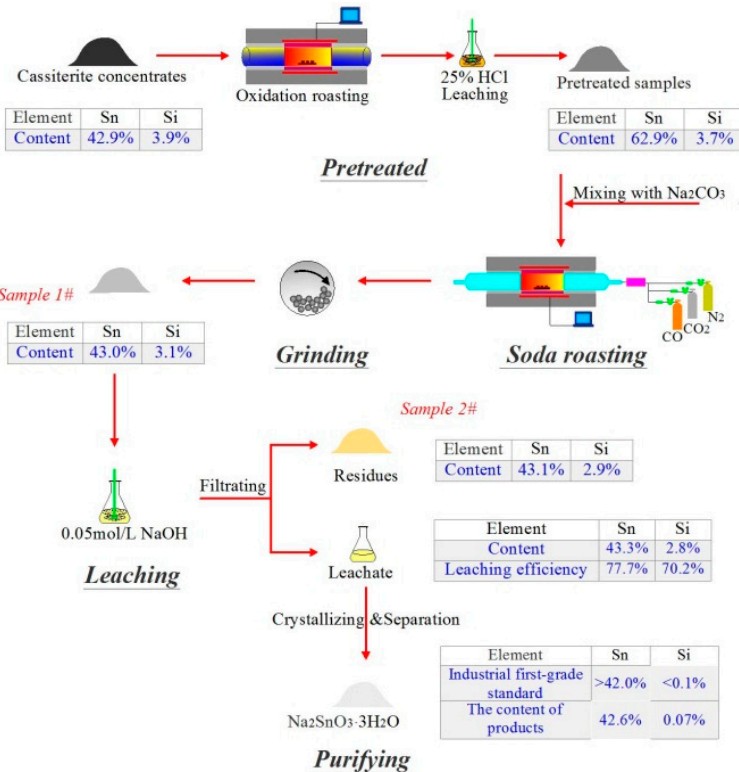

**Figure 3.** Distribution of Sn and Si elements in the whole experimental flowsheet (under the most suitable conditions).

### 3.1.2. Phase Analysis of Roasted Samples and Leach Residues

The XRD pattern raw material (in Figure 3) is shown in Figure 4, which contained tin oxide ($SnO_2$) and silicon dioxide ($SiO_2$). The XRD pattern of Sample 1# (in Figure 3) is shown in Figure 5. The main phases of the roasted samples were $Na_2SnO_3$ and $Na_2SiO_3$, and the content of Sn and Si was 43.0 wt. % and 3.1 wt. %, respectively (as shown in Figure 3). In particular, the characteristic peaks of $Na_8SnSi_6O_{18}$ were also observed in Figure 5. It was noteworthy that the phase of $Na_8SnSi_6O_{18}$ was never reported in previous researches.

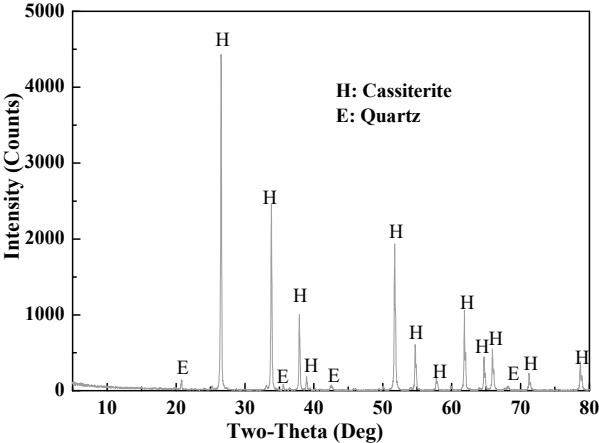

**Figure 4.** X-ray diffraction pattern of raw material.

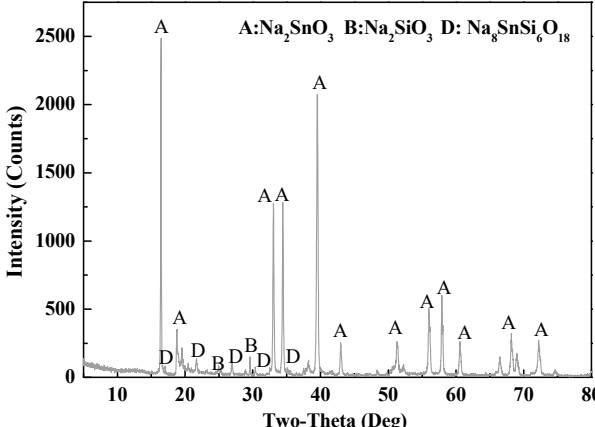

**Figure 5.** XRD pattern of the roasted samples (Sample 1# in Figure 3).

Sample 1# was then used for the leaching test. The XRD pattern of the leach residues, Sample 2# (in Figure 4), is shown in Figure 6. The content of Sn and Si in the residues was 43.1 wt. % and 2.9 wt. %, as shown in Figure 4. The main phases of the residues were $SnO_2$ and $Na_8SnSi_6O_{18}$. Moreover, the diffraction intensities of $Na_8SnSi_6O_{18}$ were more intensive compared with those shown in Figure 6, indicating that $Na_8SnSi_6O_{18}$ was enriched in the residues. Meanwhile, the diffraction peaks of $Na_2SnO_3$ and $Na_2SiO_3$ disappeared. Therefore, we drew the following conclusions: (1) a few of $SnO_2$ was were not converted into soluble stannate; (2) $Na_8SnSi_6O_{18}$ was formed in the roasting process; (3) $Na_2SnO_3$ and $Na_2SiO_3$ were almost dissolved into the leachate during the leaching process; (4) $Na_8SnSi_6O_{18}$ was possibly insoluble and enriched in the residues.

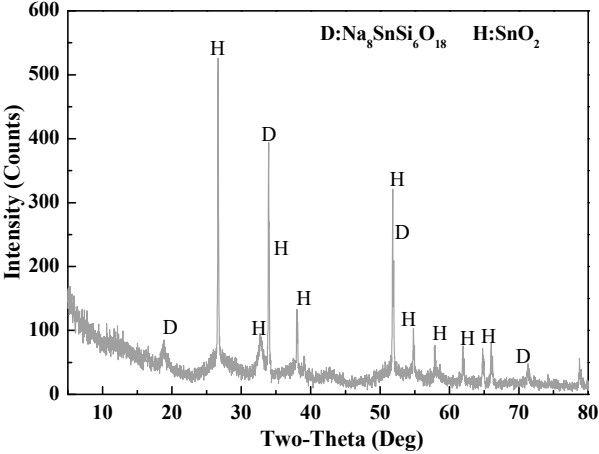

**Figure 6.** XRD pattern of the leach residues (Sample 2# in Figure 3).

### 3.1.3. Solution Chemistry of Metasilicic Acid and Tin

To further determine the existing forms of Sn and Si ions in the leachate, the solution chemistry of leachate with different pH value was analyzed. Dissolution thermodynamics of Sn and Si ions in aqueous solution was firstly discussed in the leaching process. In previous publications [35–37], it was reported that Sn(IV) might exist as $Sn^{4+}$, $SnOH^{3+}$, $Sn(OH)_2^{2+}$, $Sn(OH)_3^+$, $Sn(OH)_4$, $Sn(OH)_5^-$, and $Sn(OH)_6^{2-}$, while Si(IV) could exist as $H_2SiO_3$, $HSiO_3^-$, and $SiO_3^{2-}$. The mole fractions of metasilicic and stannum species at varying pH values were calculated based on previous studies (reaction constant (*K*) shown in Table 2) [38–41], and the results are shown in Figure 7.

**Table 2.** Reaction constants for equilibrium reactions of Si(IV) and Sn(IV) species.

| Equation | Reaction Equation | Equilibrium Constants ($K$) |
|---|---|---|
| (4) | $Sn^{4+} + H_2O = Sn(OH)^{3+} + H^+$ | $K_{\alpha1} = 10^{3.73}$ |
| (5) | $Sn^{4+} + 2H_2O = Sn(OH)_2^{2+} + 2H^+$ | $K_{\alpha1} = 10^{1.29}$ |
| (6) | $Sn^{4+} + 3H_2O = Sn(OH)_3^+ + 3H^+$ | $K_{\alpha1} = 10^{0.47}$ |
| (7) | $Sn^{4+} + 4H_2O = Sn(OH)_4 + 4H^+$ | $K_{\alpha1} = 10^{0.4}$ |
| (8) | $Sn^{4+} + 5H_2O = Sn(OH)_5^- + 5H^+$ | $K_{\alpha2} = 10^{-7.7}$ |
| (9) | $Sn^{4+} + 6H_2O = Sn(OH)_6^{2-} + 6H^+$ | $K_{\alpha3} = 10^{-18.1}$ |
| (10) | $SiO_3^{2-} + H+ = HSiO_3^-$ | $K_{\beta1} = 10^{-11.82}$ |
| (11) | $HSiO_3^- + H+ = H_2SiO_3$ | $K_{\beta2} = 10^{-9.69}$ |

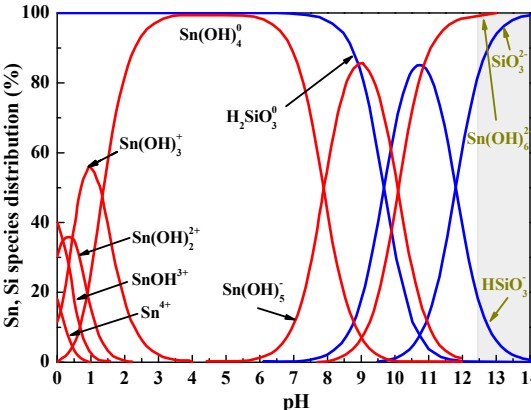

**Figure 7.** Mole fractions of metasilicic and stannum species at different pH values.

Figure 7 illustrated the very strong hydrolysis behaviors of Si(IV) and Sn(IV). The main species of metasilicic and stannum were $Sn^{4+}$, $SnOH^{3+}$, $Sn(OH)_2^{2+}$, and $H_2SiO_3$ at pH value below 2. The neutral $Sn(OH)_4$ and $H_2SiO_3$ were dominant at pH value of 2–7.5. At pH of 9.5, the negatively charged $HSiO_3^-$, $SiO_3^{2-}$, $Sn(OH)_5^-$, and $Sn(OH)_6^{2-}$ were the main aqueous species. However, after pH value above 12.5, the mole fraction of $HSiO_3^-$ was lower compared to that of $SiO_3^{2-}$. Moreover, $Sn(OH)_4$ and $H_2SiO_3$ disappeared. Generally, a leaching solvent with a pH greater than 12 is preferred for the prevention of stannate hydrolysis [14]. Hence, $Sn(OH)_6^{2-}$ and $SiO_3^{2-}$ only existed in the solution when the value of pH was more than 12.6 (0.05 mol/L NaOH). Figure 6 also indicated that there were no $H_2SiO_3(s)$ and $Sn(OH)_4(s)$ in the leach residues, illustrating that $H_2SiO_3(s)$ and $Sn(OH)_4(s)$ was not formed during the leaching process.

The possible reaction of $Na_2SnO_3(s)$ during the leaching process was expressed as Equation (12), and $Sn(OH)_6^{2-}$ anions were instantly formed because of their high stability under weakly alkaline condition [42]. $Na_2SiO_3(s)$ was dissolved as forms of $Na^+$ and $SiO_3^{2-}$ (aq.) by Equation (13) at pH of 12.6 [36]. However, $Sn(OH)_6^{2-}$ did not react with $SiO_3^{2-}$ to form $Na_8SnSi_6O_{18}$ in the solution with pH value of 12.6. Therefore, we inferred that $Na_8SnSi_6O_{18}$ was only formed in the roasting process.

$$Na_2SnO_3(s) + 3H_2O = Na_2Sn(OH)_6 \text{ (aq.)} = 2Na^+ + Sn(OH)_6^{2-} \tag{12}$$

$$Na_2SiO_3(s) = 2Na^+ + SiO_3(aq)^{2-} \tag{13}$$

The abovementioned results indicated that $SiO_2$ in the cassiterite concentrates could react with $Na_2CO_3$ and $SnO_2$ to form $Na_2SnO_3$, $Na_2SiO_3$, and $Na_8SnSi_6O_{18}$ during the roasting process. Part of $SiO_2$ reacted with $SnO_2$ and $Na_2CO_3$ to form $Na_8SnSi_6O_{18}$. Moreover, $Na_2SnO_3$ and $Na_2SiO_3$ were more easily soluble than $Na_8SnSi_6O_{18}$. In order to make sure how $Na_8SnSi_6O_{18}$ was formed during the roasting process, effect of $SiO_2$ on phase evolution of $SnO_2$–$Na_2CO_3$ system was further researched.

### 3.2. Effect of SiO₂ on Phase Evolution of SnO₂–Na₂CO₃ System

It was reported that $SiO_2/SnO_2$ molar ratio in cassiterite concentrates was about 1:4 [14]. Hence, in order to research the effect of $SiO_2$ on phase evolution of $SnO_2$–$Na_2CO_3$ system, the analytical grade reagents of $Na_2CO_3$ and $SiO_2$ were added into cassiterite concentrates with different $SiO_2/SnO_2$ molar ratios. In particular, $Na_2CO_3/(SnO_2 + SiO_2)$ mole ratio was fixed as 1.5.

### 3.2.1. Effect of SiO₂/SnO₂ Mole Ratio

Firstly, different $SiO_2$ dosage was added into cassiterite concentrates in order to investigate the effect of $SiO_2/SnO_2$ mole ratio. The XRD patterns of the samples roasted at 875 °C for 60 min in a 15 vol. % CO atmosphere are shown in Figure 8. $SiO_2/SnO_2$ mole ratio varied in the range of 1:4–7:1.

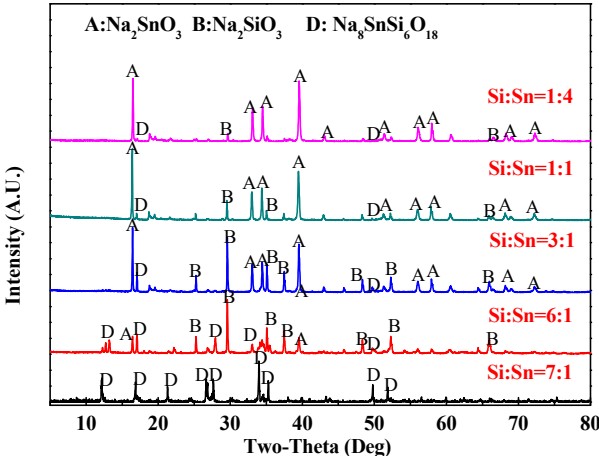

**Figure 8.** XRD patterns of the samples with different $SiO_2/SnO_2$ mole ratio (Temperature: 875 °C; Time: 60 min).

As observed from Figure 8, the main phases of the roasted samples were $Na_2SnO_3$, $Na_2SiO_3$, and a small amount of $Na_8SnSi_6O_{18}$ when the $SiO_2/SnO_2$ mole ratio was 1:4. As the $SiO_2/SnO_2$ mole ratio increased from 1:4 to 7:1, the diffraction peak intensity of $Na_8SnSi_6O_{18}$ increased significantly. Meanwhile, the diffraction peak of $Na_2SnO_3$ was gradually weakened and vanished. The Sn/Si theoretical value in $Na_8SnSi_6O_{18}$ was 1:6. The results indicated that high Si content in cassiterite concentrates promoted the formation of $Na_8SnSi_6O_{18}$, which decreased the conversion of $SnO_2$ to $Na_2SnO_3$ during the roasting process.

### 3.2.2. Effect of Roasting Temperature

The effect of roasting temperature was then performed, and Figure 9 shows the XRD patterns of the samples roasted for 60 min at temperature range of 775–925 °C. $SiO_2/SnO_2$ mole ratio was fixed as 7:1.

It was found that the phases of the samples roasted at 775 °C were $Na_8SnSi_6O_{18}$, $Na_2SiO_3$, and a small amount of $Na_2CO_3$. The diffraction peak intensities of $Na_8SnSi_6O_{18}$ were enhanced with increasing the roasting temperature, while those of $Na_2SiO_3$ and $Na_2CO_3$ were weakened. When the temperature reached at 875 °C, $Na_8SnSi_6O_{18}$ was the predominant substance in the roasted samples. The diffraction peaks of $Na_2SiO_3$ and $Na_2CO_3$ almost vanished as the roasting temperature increased to 925 °C, and $Na_8SnSi_6O_{18}$ was the only phase in the roasted samples.

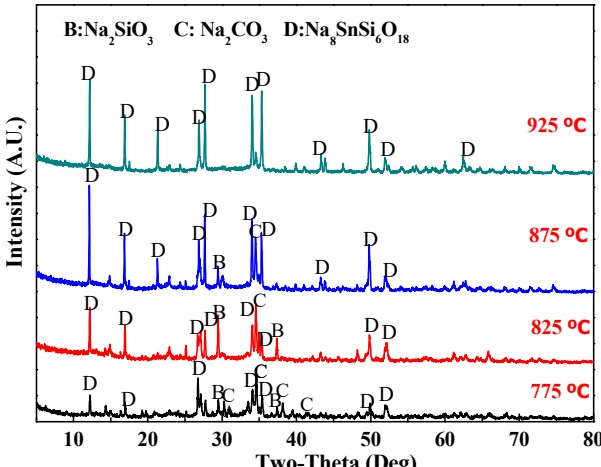

**Figure 9.** XRD patterns of the samples roasted at different temperatures (Time: 60 min; $SiO_2/SnO_2$ molar ratio: 7:1).

It was worthy to note that 875 °C was the suitable roasting temperature for $Na_2SnO_3$ preparation by the soda roasting–leaching process [14]. Besides, as observed from Figure 4, the leaching efficiency of Sn and Si at 875 °C was 77.7 wt. % and 70.2 wt. %, respectively. Furthermore, the leaching efficiency of Sn and Si was derived from the formation of soluble $Na_2SnO_3$ and $Na_2SiO_3$. The results further illustrated that a part of $SnO_2$ and $SiO_2$ was transformed into $Na_8SnSi_6O_{18}$. The corresponding reaction was expressed as the reaction of Equation (14).

$$SnO_2 + 6SiO_2 + 4Na_2CO_3 = Na_8SnSi_6O_{18} + 4CO_2 \tag{14}$$

### 3.3. Leaching Behavior of $Na_8SnSi_6O_{18}$

$Na_8SnSi_6O_{18}$ was inevitably formed in the roasting process, and it was enriched in the leach residues. However, the leaching behavior of $Na_8SnSi_6O_{18}$ was never conducted. In this section, to determine the leaching behavior of $Na_8SnSi_6O_{18}$, $Na_8SnSi_6O_{18}$ was synthesized under the following conditions: mole ratio of $Na_2CO_3$:$SnO_2$:$SiO_2$ = 4:1:6, roasting temperature of 1000 °C, and roasting time of 120 min. The XRD pattern of synthetic $Na_8SnSi_6O_{18}$ is shown in Figure 10. The results indicated that the roasted product had a very high purity and was well-matched with the PDF standard card (PDF#72-2449) of $Na_8SnSi_6O_{18}$ phase, and there were no diffraction peaks of other impurities.

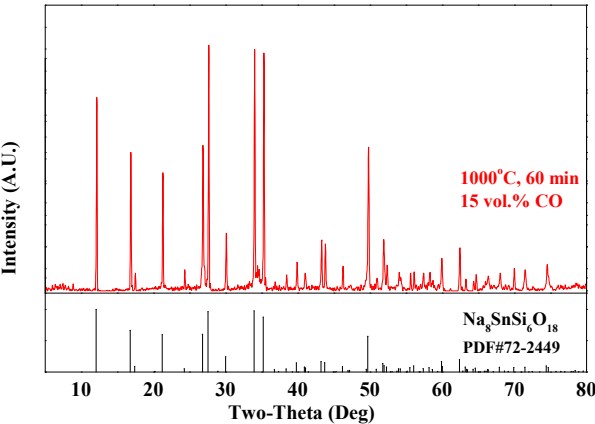

**Figure 10.** XRD pattern of the synthetic $Na_8SnSi_6O_{18}$.

Effect of pH on leaching behavior of $Na_8SnSi_6O_{18}$ was performed under the following leaching experimental conditions: liquid-to-solid ratio of 4 $cm^3$/g, leaching temperature of 40 °C, leaching time of 60 min, and stirring rate of 300 rpm [14].

The effect of pH value on the leaching efficiency of Si and Sn was investigated systematically, and the results are shown in Figure 11a. The leaching efficiency of Sn decreased from 65.1 wt. % to 1.9 wt. % and Si from 21.2 wt. % to 4.9 wt. % as the pH value increased from −0.6 (4 mol/L HCl) to 1. The leaching efficiency of Si and Sn was substantially unchanged as the pH value increased further. The leaching efficiency of Sn and Si was 0.5 wt. % and 1.3 wt. % (in Figure 11a) when the pH value was 12.6 (0.05 mol/L NaOH). The XRD pattern of the leach residues obtained at pH of 12.6 is also presented in Figure 11b. It is seen from Figure 11b that $Na_8SnSi_6O_{18}$ was the only phase in the residues. The results verified that $Na_8SnSi_6O_{18}$ phase was almost insoluble and residual in the residues.

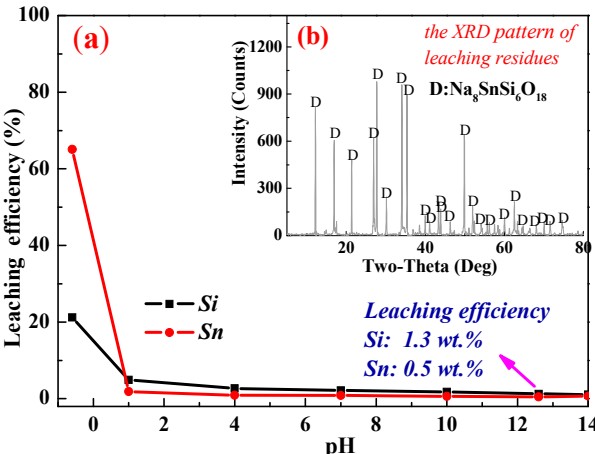

**Figure 11.** The leaching efficiency Si and Sn of the synthetic $Na_8SnSi_6O_{18}$. (**a**) leaching efficiency Si and Sn, (**b**) XRD pattern of leach residues.

In the whole, $Na_8SnSi_6O_{18}$ was inevitably formed in the roasting process for preparation of $Na_2SnO_3$ if using cassiterite as raw materials. Moreover, $Na_8SnSi_6O_{18}$ was insoluble in the pH range of 1–14, which was the main reason that part of tin oxides was not converted into soluble $Na_2SnO_3$ and enriched in the residues. Thus, if cassiterite concentrates were used as raw materials to prepare $Na_2SnO_3$ by the soda roasting–leaching process, the silicon content in the cassiterite should be controlled as low as possible.

## 4. Conclusions

In this study, the effect of $SiO_2$ on sodium stannate preparation from cassiterite and $Na_2CO_3$ roasted under CO–CO$_2$ atmosphere was investigated. The main conclusions were summarized as follows:

1.  $Na_2SnO_3$, $Na_2SiO_3$, and $Na_8SnSi_6O_{18}$ were easily formed when the $Na_2CO_3$ + $SnO_2$ + $SiO_2$ mixtures were roasted under CO–CO$_2$ atmosphere. $Na_2SnO_3$ and $Na_2SiO_3$ were easily dissolved into the leachate during the leaching process, while $Na_8SnSi_6O_{18}$ enriched into the leach residues.
2.  $Na_8SnSi_6O_{18}$ was inevitably formed in the roasting process of preparation of $Na_2SnO_3$. Roasting temperature and Si/Sn mole ratio were the two critical factors affecting the formation of $Na_8SnSi_6O_{18}$, which was more easily formed at higher roasting temperature and Si/Sn mole ratio.
3.  The leaching behavior of synthetic $Na_8SnSi_6O_{18}$ indicated that $Na_8SnSi_6O_{18}$ was almost insoluble in the leachate at the pH range of 1–14. Therefore, the loss of tin in the residues was mainly attributed to the insoluble $Na_8SnSi_6O_{18}$ formed during the roasting process.

**Author Contributions:** Y.Z. conceived the project and wrote the final paper. Z.S. performed the experiments and wrote initial drafts of the work. B.H. performed the experiment. X.C. performed the XRD analysis. M.L., S.L., J.L., and T.J. discussed the content. All authors discussed the results and reviewed the manuscript.

**Funding:** This research was funded by National Natural Science Foundation of China (No.51574283 and No.51234008).

**Acknowledgments:** The authors would express their heartful thanks to Financial supports from the National Natural Science Foundation of China and Co-Innovation Center for Clean and Efficient Utilization of Strategic Metal Mineral Resources, and language editing assistance from Corby Anderson in Kroll Institute for Extractive Metallurgy, Colorado School of Mines, USA.

**Conflicts of Interest:** The authors declare no conflict of interest.

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
