# Peer review of "Effect of Quartz on the Preparation of Sodium Stannate from Cassiterite Concentrates by Soda Roasting Process"

_minerals, doi:10.3390/min9100605_

Round 1

Reviewer 1 Report

Peer Review Report: “Effect of Quartz on the Preparation of Sodium Stannate from Cassiterite Concentrates by Soda Roasting Process

Comments to the Authors.

The paper describes a novel process for the recovery of tin from  cassiterite using a novel methodology based on alkali roasting with Na2CO3 in a reducing atmosphere. The reviewer thinks that this paper is suitable for publication in the journal Minerals after minor corrections .

Minor comments.

1. The paper needs to be revised to correct minor English language issues:

          Page 2 Line 48 “can´t” should be replace by cannot

          Page 2 Line 69 “The materials were weighed precisely as a certain                  molar ratio” should be “The materials were weighed precisely at a                  certain molar ratio”

         Page 6 Line 166 “3.1.3. Solution chemistry of metasilicic and stannum”           should be “3.1.3. Solution chemistry of metasilicic acid and tin”

         Page 7 Line 213 “which decreased the convert ratio of SnO2 to” should           be “which decreased the conversion of SnO2 to”

2. In the introduction the authors should summarized the main differences between the oxidative and the reductive alkali roasting processes, to highlight the advantages of the novel process.

3. The full chemical composition of the cassiterite concentrate should be included in the manuscript.

4. In Figure 3 a mass balance should be included. Is any Sn lost after the 25%HCl leaching?

5. The XRD pattern of the leaching residue in Figure 6 shows the presence of SnO2. Is this attributed to a lack of Na2CO3 in the roasting process?

6. Is the shaded area in Figure 7, the pH region in which the leaching process takes place? If so, please clarify it in the manuscript.

7. I recommend the authors to read the following references and include them if appropriate:

Sanchez-Segado, S., Monti, T., Katrib, J., Kingman, S., Dodds, C., Jha,            A. Towards sustainable processing of columbite group minerals:   Elucidating the relation between dielectric properties and physico-chemical transformations in the mineral phase (2017) Scientific Reports, 7 (1), art. no. 18016.

Escudero-Castejon, L., Sanchez-Segado, S., Parirenyatwa, S., Hara, Y., Jha, A. A Cr6+-free extraction of chromium oxide from chromite ores using carbothermic reduction in the presence of alkali(2017) Minerals, Metals and Materials Series, pp. 179-188.

Parirenyatwa, S., Escudero-Castejon, L., Sanchez-Segado, S., Hara, Y., Jha, A. An investigation on the kinetics and mechanism of alkali reduction of mine waste containing titaniferous minerals for the recovery of metals(2017) Minerals, Metals and Materials Series, (9783319510903), pp. 465-474.

Sanchez-Segado, S., Makanyire, T., Escudero-Castejon, L., Hara, Y., Jha, A. Reclamation of reactive metal oxides from complex minerals using alkali roasting and leaching - An improved approach to process engineering(2015) Green Chemistry, 17 (4), pp. 2059-2080.

Sanchez-Segado, S., Lahiri, A., Jha, A. Alkali roasting of bomar ilmenite: Rare earths recovery and physico-chemical changes(2015) Open Chemistry, 13 (1), pp. 270-278.

Author Response

Response to Reviewer 1 Comments

The paper needs to be revised to correct minor English language issues:

Page 2 Line 48 “can´t” should be replace by cannot

Page 2 Line 69 “The materials were weighed precisely as a certain molar ratio” should be “The materials were weighed precisely at a certain molar ratio”

Page 6 Line 166 “3.1.3. Solution chemistry of metasilicic and stannum” should be “3.1.3. Solution chemistry of metasilicic acid and tin”

Page 7 Line 213 “which decreased the convert ratio of SnO2 to” should be “which decreased the conversion of SnO2 to”

Response: Thanks for your suggestion. We have revised the language issues, in order to make the changes easily viewable for you, we marked the revisions with red colour.

In the introduction the authors should summarized the main differences between the oxidative and the reductive alkali roasting processes, to highlight the advantages of the novel process.

Response : Our previous studies have discussed in detail about the formation mechanism of Na2SnO3 in reductive atmosphere ([1] Zhang, Y.; Su, Z.; Liu, B.; You, Z.; Yang, G.; Li, G.; Jiang, T. Sodium stannate preparation from stannic oxide by a novel soda roasting-leaching process. Hydrometallurgy 2014, 146, 82-88. [2] Liu, B.; Zhang, Y.; Su, Z.; Li, G.; Jiang, T. Phase Evolution of Tin Oxides Roasted Under CO-CO2 Atmospheres in the Presence of Na2CO3. Mineral Processing and Extractive Metallurgy Review 2016, 37, 264-273. [3] Liu, B.; Zhang, Y.; Su, Z.; Li, G.; Jiang, T. Function mechanism of CO-CO2 atmosphere on the formation of Na2SnO3 from SnO2 and Na2CO3 during the roasting process. Powder Technology 2016, 301, 102-109. [4] Liu, B.; Zhang, Y.; Su, Z.; Li, G.; Jiang, T. Formation kinetics of Na2SnO3 from SnO2 and Na2CO3 roasted under CO-CO2 atmosphere. International Journal of Mineral Processing 2017, 165, 34-40). However the major objective of this study were to study the effect of quartz on the preparation of sodium stannate from cassiterite concentrates by soda roasting process.

The full chemical composition of the cassiterite concentrate should be included in the manuscript.

Response : Our previous studies showed that trihydrate sodium stannate (Na2SnO3·3H2O) was obtained by roasting cassiterite concentrates and Na2CO3 in CO-CO2 atmosphere (Zhang, Y.; Su, Z.; Liu, B.; You, Z.; Yang, G.; Li, G.; Jiang, T. Sodium stannate preparation from stannic oxide by a novel soda roasting-leaching process. Hydrometallurgy 2014, 146, 82-88). The full chemical composition of the cassiterite concentrate and products has been given in that reference. And the table of chemical composition was added in the revised manuscript.

In Figure 3 a mass balance should be included. Is any Sn lost after the 25%HCl leaching?

Response : Tin dioxide was insoluble in 25% HCl leaching, so the loss of Sn can be neglected.

The XRD pattern of the leaching residue in Figure 6 shows the presence of SnO2. Is this attributed to a lack of Na2CO3 in the roasting process?

Response: Our studies have found that Na8SnSi6O18 was a low melting point phase, it may wrapped rapidly on the surface of cassiterite concentrate, resulting unreacted cassiterite remained in the leaching residues. So, this part of SnO2 would be further study in the future. .

Is the shaded area in Figure 7, the pH region in which the leaching process takes place? If so, please clarify it in the manuscript.

Response: We have clarified it in the manuscript as follows: Generally, a leaching solvent with a pH greater than 12 is preferred for the prevention of stannate hydrolysis (Zhang, Y.; Su, Z.; Liu, B.; You, Z.; Yang, G.; Li, G.; Jiang, T. Sodium stannate preparation from stannic oxide by a novel soda roasting-leaching process. Hydrometallurgy 2014, 146, 82-88). Hence, Sn(OH)62- and SiO32- only existed in the solution when the value of pH was more than 12.6 (0.05 mol/L NaOH).

I recommend the authors to read the following references and include them if appropriate: Sanchez-Segado, S., Monti, T., Katrib, J., Kingman, S., Dodds, C., Jha, A. Towards sustainable processing of columbite group minerals: Elucidating the relation between dielectric properties and physico-chemical transformations in the mineral phase. Scientific Reports 2017, 7 (1), art. no. 18016.

Escudero-Castejon, L., Sanchez-Segado, S., Parirenyatwa, S., Hara, Y., Jha, A. A Cr6+-free extraction of chromium oxide from chromite ores using carbothermic reduction in the presence of alkali. Minerals, Metals and Materials Series 2017, pp. 179-188.

Parirenyatwa, S., Escudero-Castejon, L., Sanchez-Segado, S., Hara, Y., Jha, A. An investigation on the kinetics and mechanism of alkali reduction of mine waste containing titaniferous minerals for the recovery of metals. Minerals, Metals and Materials Series 2017, (9783319510903), pp. 465-474.

Sanchez-Segado, S., Makanyire, T., Escudero-Castejon, L., Hara, Y., Jha, A. Reclamation of reactive metal oxides from complex minerals using alkali roasting and leaching - An improved approach to process engineering(2015) Green Chemistry, 17 (4), pp. 2059-2080.

Response : Thanks for your suggestion. We have added the following references in the revised manuscript.

Reviewer 2 Report

The manuscript is concerned with the effect of SiO2 on the formation of sodium stannate from cassiterite concentrates by soda roasting process. The manuscript is concerned mostly with the presentation of experimental results since it is part of a series of publications on the subject. However, because of that, it is difficult to follow some of the discussions in this manuscript because there is a lack of detailed information on the experimental conditions of the work presented here. Nevertheless, the paper could be of interest to readers working in the lime soda roasting process.

The deficiencies of the manuscript that must be take care of are listed below.

-          In line 55-56, the second objective stated in this manuscript is the determination of the effect of SiO2 on the leaching efficiency of Sn and Si. However, the leaching variables were not studied experimentally, only the roasting conditions were varied. The leaching efficiency L, as defined by the authors in line 89, is nothing but the percent dissolution of Sn or Si from soluble compounds formed in the soda roasting process, and thus, the ordinate of Figures 1 and 2 are misleading.

-          For the same reason, the subheading 3.1, (Line 101) is also misleading and should be changed.  

-          The authors claimed (in line 114) that the leaching efficiency increased significantly with an increase in the roasting temperatures.  In addition, at 1000 °C the leaching efficiency was the same for Sn and Si.  This is not true, actually what increased with an increase in temperature is the extent of formation of separate soluble compounds of Sn or Si or a complex Sn-Si compounds in roasting.  These needs a better explanation.

               In line 20, change accelerated to increased?

                In line 2, faction to fraction?

                In line 64, include sodium carbonate

                In line 69, which materials?

                In line 85, “leached at different pH solutions for 60 min”. Which were the pH’s used? What     were the pH used for the data shown in Fig. 1 and 2?

Author Response

Response to Reviewer 2 Comments

Point 1: The deficiencies of the manuscript that must be take care of are listed below.

In line 55-56, the second objective stated in this manuscript is the determination of the effect of SiO2 on the leaching efficiency of Sn and Si. However, the leaching variables were not studied experimentally, only the roasting conditions were varied. The leaching efficiency L, as defined by the authors in line 89, is nothing but the percent dissolution of Sn or Si from soluble compounds formed in the soda roasting process, and thus, the ordinate of Figures 1 and 2 are misleading.

Response 1: Based on our previous studies, the effect of leaching parameters were investigated systematically [14]. Fig. 1 and Fig. 2 were just to present the leaching correlation efficiency of Sn and Si, which to observe the effect of Si on the formation of Na2SnO3. Hence, we think this paper were necessary in this paper.

Point 2: For the same reason, the subheading 3.1, (Line 101) is also misleading and should be changed.

Response 2: Thanks for your suggestion, we have modified it.

Point 3: The authors claimed (in line 114) that the leaching efficiency increased significantly with an increase in the roasting temperatures.  In addition, at 1000 °C the leaching efficiency was the same for Sn and Si.  This is not true, actually what increased with an increase in temperature is the extent of formation of separate soluble compounds of Sn or Si or a complex Sn-Si compounds in roasting.  These needs a better explanation.

Response 3: Thanks for your good suggestion, we have checked the results carefully and they were correct. During the soda roasting process, Na2CO3 content was excess. And Sn-Si compounds will further react with Na2CO3 at a higher temperature. This reaction mechanism will be discussed in our future papers.

Point 4: In line 20, change accelerated to increased? In line 2, faction to fraction? In line 64, include sodium carbonate? In line 69, which materials?

Response 4: We have modified those mistakes in the revised manuscripts.

Point 5: In line 85, “leached at different pH solutions for 60 min”. Which were the pH’s used? What  were the pH used for the data shown in Fig. 1 and 2?

Response 5: The pH was fixed at 12.5 based on our previous study [14].